# Time Spent with Saturation below 80% versus 90% in Patients with Obstructive Sleep Apnoea

**DOI:** 10.3390/jcm12134205

**Published:** 2023-06-22

**Authors:** András Bikov, Stefan Frent, Oana Deleanu, Martina Meszaros, Mariela Romina Birza, Alina Mirela Popa, Andrei Raul Manzur, Loredana Gligor, Stefan Mihaicuta

**Affiliations:** 1Wythenshawe Hospital, Manchester University NHS Foundation Trust, Manchester Academic Health Science Centre, Oxford Road, Manchester M13 9WL, UK; andras.bikov@gmail.com; 2Division of Infection, Immunity & Respiratory Medicine, Faculty of Biology, Medicine and Health, The University of Manchester, Manchester M13 9PT, UK; 3Center for Research and Innovation in Precision Medicine of Respiratory Diseases, Department of Pulmonology, “Victor Babes” University of Medicine and Pharmacy Timisoara, Eftimie Murgu Sq. No. 2, 300041 Timisoara, Romania; frentz.stefan@umft.ro (S.F.); ally.mirela@yahoo.com (A.M.P.); andrei.manzur.raul@gmail.com (A.R.M.); gligorloredananeli@gmail.com (L.G.); stefan.mihaicuta@umft.ro (S.M.); 4Department of Pulmonology, University of Medicine and Pharmacy Carol Davila, Bulevardul Eroii Sanitari 8, 050474 Bucharest, Romania; oanadeleanu@gmail.com; 5Department of Pulmonology, Semmelweis University, Tömő Street 25-29, Budapest 1083, Hungary; martina.meszaros1015@gmail.com

**Keywords:** hypoxemia, cardiovascular risk, inflammation, burden

## Abstract

Background: Nocturnal hypoxaemia measured as the percentage of total sleep time spent with saturation below 90% (TST90%) may better predict cardiovascular consequences of obstructive sleep apnoea (OSA) than the number of obstructive respiratory events measured with the apnoea–hypopnea index (AHI). Deeper hypoxaemia may potentially induce more severe pathophysiological consequences. However, the additional value of the percentage of total sleep time spent with saturation below 80% (TST80%) to TST90% is not fully explored. Methods: Comprehensive medical history was taken and fasting lipid and C-reactive protein levels were measured in 797 volunteers participating in two cohort studies in Hungary and Romania. Sleep parameters, including AHI, TST90% and TST80%, were recorded following a polysomnography (PSG, *n* = 598) or an inpatient cardiorespiratory polygraphy (*n* = 199). The performance of TST80% to predict cardiovascular risk was compared with TST90% using linear and logistic regression analyses as well receiver operating characteristics curves. Sensitivity analyses were performed in patients who had PSG, separately. Results: Both parameters are significantly related to cardiovascular risk factors; however, TST80% did not show better predictive value for cardiovascular risk than TST90%. On the other hand, patients with more severe hypoxaemia reported more excessive daytime sleepiness. Conclusions: TST80% has limited additional clinical value compared to TST90% when evaluating cardiovascular risk in patients with OSA.

## 1. Introduction

Obstructive sleep apnoea (OSA) is a common condition characterised by repetitive partial or total collapse of the upper airways during sleep. It leads to night- and daytime symptoms and is a risk factor for the development of cardiovascular disease. However, whilst treatment with positive airway pressure (PAP) has been shown to improve symptoms associated with OSA, it has failed to demonstrate adequate protection from cardiovascular morbidity and mortality in randomised controlled trials [1,2,3,4]. A potential reason for this could be that the mechanisms linking OSA to cardiovascular disease are not fully understood.

High blood pressure, diabetes, dyslipidaemia and systemic inflammation are all associated with OSA [5,6,7,8] and are independent risk factors for cardiovascular disease [9]. Chronic intermittent hypoxaemia is the main mechanisms leading to their development in OSA [8]. It has recently been recognised that sleep-related hypoxaemia, measured according to the time spent with saturation below 90% (TST90%), more strongly correlates with cardiovascular events than the number of obstructive respiratory events, measured using the apnoea–hypopnoea index (AHI) [10,11,12,13]. Whilst the severity of OSA is traditionally based on discrete AHI cut-off values (i.e., AHI 5–14.9/h is mild, AHI 15–29.9/h is moderate and AHI ≥ 30 is severe OSA), no such classification exists for TST90% values. A previous study introduced light (TST90% ≤ 5%), mild (TST90% 5–10%), moderate (TST90% 10–25%) and severe (TST90% > 25%) categories. This classification was able to stratify hypertension risk in patients with OSA [14].

Hypoxaemia exerts its effect through hypoxia-inducible factors (HIF). These transcriptional factors are activated by low oxygen tension, affect the cellular metabolism and induce an inflammatory response [15]. The role of HIF-1α has been implicated in both dyslipidaemia and diabetes [5,16]. In addition, HIF-1α contributes to the augmented chemosensory reflex leading to the development of hypertension [17]. The activity of HIF-1α directly correlates with the severity of hypoxaemia [18]. Mice exposed to increasing levels of intermittent hypoxaemia exhibited hypoxaemia-dependent hypertriglyceridaemia [19,20]. In addition, the carotid body chemosensory response is correlated with the level of hypoxaemia [21] and the hypoxaemia-induced increase in adrenalin and noradrenalin is also inversely related to oxygen tension [20]. Similarly, glucose levels are related to the intensity of hypoxaemia, but this mechanism is strongly modulated by the sympathetic tone [22]. The experimental studies above suggest that deeper hypoxaemia could be more harmful, and the level of hypoxaemia could potentially add further clinical information to predict cardiovascular disease in OSA compared to the overall time spent in nocturnal hypoxaemia.

There are several markers to measure hypoxaemic burden during sleep, including minimal saturation (MinSatO_2_), the oxygen desaturation index (ODI), TST90%, sleep-apnoea-specific hypoxic burden (measured as the area under the desaturation curve associated with respiratory events) [23,24], desaturation duration (the cumulative time of desaturations) and desaturation severity (the sum of the areas of all desaturation events) [25,26]. Whilst the former three are readily available on sleep reports, the latter three need to be calculated with external software. The merit of the latter indices is that they take into account both the depth and duration of desaturation; however, their additive value compared to TST90% is limited [23,25] but might outperform TST90% in selected populations [23]. Time spent with saturation below 80% (TST80%) is a metric which is also often readily provided in sleep studies and does not require further transformation. It is independent from manual scoring, and may need only a review to exclude artefacts. TST80% comprises both desaturations’ depth and their duration. In a large cohort of community-dwelling older men, in contrast to TST90%, TST80% > 1% was an independent risk factor for all-cause mortality [11]. On the other hand, in the Sleep Heart Health Study, TST90% but not TST80% was related to incidental cardiovascular disease [27]. The discrepancy could be due to the different burden of cardiovascular risk factors, such as blood lipids or hypertension, which were not reported in these studies.

Our aim was to specifically investigate the relationship between TST80% and cardiovascular risk in OSA. The primary aim was to compare its value with TST90% to predict hypertension, diabetes, dyslipidaemia, cardiovascular disease and enhanced inflammatory response in patients with OSA.

## 2. Methods

### 2.1. Design and Subjects

We analysed the results of 2679 volunteers participating in two prospective cohort studies from Budapest, Hungary and Timisoara, Romania. In both cohorts, subjects were referred with suspected diagnosis of sleep apnoea based on symptoms (i.e., snoring, witnessed pauses in breathing, daytime tiredness) and comorbidities. None of the volunteers were diagnosed or treated for OSA before. We excluded patients with missing medical history; patients with chronic lung, chest or neuromuscular disease that could lead to chronic overnight hypoxaemia; if the sleep study data were no longer available or if patients did not have a blood lipid profile. As a result, we studied the results of 797 volunteers in the final analysis (Figure 1).

Hypertension, diabetes and cardiovascular disease (stable angina, previous cardiovascular or cerebrovascular event) were defined based on medical history and relevant medications. Dyslipidaemia was defined based on the National Cholesterol Education Programs Adult Treatment Panel III criteria as triglycerides levels ≥  1.7 mmol/L and/or high-density lipoprotein cholesterol (HDL-C) levels  <  1.03 mmol/L in men or < 1.29 mmol/L in women or lipid-lowering treatment. Patients were defined smokers if they were currently smoking or gave up within 12 months [28]. Venous blood samples were taken in fasting conditions for measuring the level of total cholesterol, low-density lipoprotein cholesterol (LDL-C), HDL-C, triglycerides and C-reactive protein (CRP). In addition, morning blood pressures following the sleep studies were recorded and the participants filled out the Epworth Sleepiness Scale (ESS). We have also calculated the Framingham risk score, which is a sex-specific algorithm to calculate 10-year cardiovascular risk. The risk is calculated based on age, sex, current smoking, blood pressure value (and if the patient on an anti-hypertensive medication), total cholesterol and HDL-C values [28].

### 2.2. Sleep Studies

Inpatient polysomnography (PSG, *n* = 598) and cardiorespiratory polygraphy (*n* = 199) were performed according to the recommendations of the American Academy of Sleep Medicine (AASM) [29]. The choice of the test was based on pre-test likelihood of OSA and the presence of complicating factors as suggested by the AASM [30]. Sleep stages and cardiorespiratory events were manually scored according to the AASM guidelines [31]. Apnoea was defined as at least 90% drop in the nasal airflow lasting for at least 10 s. Hypopnoea was defined as at least 30% drop in the nasal airflow that was accompanied by at least 3% drop in the oxygen levels (and/or arousal on PSG). We recorded AHI, ODI, TST80% and TST90%. An AHI ≥ 5/h was diagnostic for OSA. TST80% and TST90% were expressed as percentages of total sleep time. Based on the sleep studies, the population was divided into control (no OSA, group 1), non-hypoxaemic OSA (AHI ≥ 5/h, but TST90% = 0%, group 2), minimally hypoxaemic OSA (AHI ≥ 5/h, TST90% 0–10%, group 3), moderately hypoxaemic OSA (AHI ≥ 5/h, TST90% ≥ 10%, but TST80% < 10%, group 4) and severely hypoxaemic OSA (AHI ≥ 5/h, and TST80% ≥ 10%, group 5) groups. The 10% TST90% cut off for the severity of hypoxaemia was based on a recent report [14].

### 2.3. Statistical Analyses

JASP 0.14.1 (University of Amsterdam, Amsterdam, The Netherlands) and SPSS 25 (IBM, New York, NY, USA) were used for statistical analysis. The groups were compared with Chi-square and ANOVA tests; the latter was followed by Tukey post hoc tests between each group. Correlations between clinical variables and TST80% as well as TST90% were investigated with Spearman’s test. Receiver operating characteristics (ROC) curves were plotted and areas under the curves (AUCs) were calculated to assess whether AHI, TST90% or TST80% predicted hypertension, diabetes, dyslipidaemia and cardiovascular disease better. As a sensitivity analysis, groups 4 and 5 were also compared following adjustment for age, gender, BMI, smoking status, lipid-lowering medication use and AHI using ANCOVA and multivariate logistic regression analyses. Further sensitivity analyses were performed in patients who had PSG as a diagnostic test. Data are expressed as mean ± standard deviation, AUC are expressed as mean/95% confidence intervals/. A *p* value < 0.05 was considered significant.

## 3. Results

### 3.1. Subjects’ Characteristics

There were significant differences in each variable among the groups (Table 1).

Most particularly, there was a progressive increase in age with worsening degrees of overnight hypoxaemia; however, interestingly, patients in group 5 were significantly younger than in group 4. Body mass index (BMI), systolic blood pressure (SBP) and diastolic blood pressure (DBP) values as well as triglyceride levels progressively increased from group 1 to group 4, but there was no difference between group 4 and group 5. LDL-C concentrations were significantly higher only in group 5 compared to group 1. HDL-C levels were lower in all OSA groups compared to group 1, without any difference between the OSA subgroups. In contrast, there was no difference between the groups in total cholesterol, whilst, surprisingly, CRP levels were lower in group 3. The Framingham risk score was higher in all OSA groups, but there was no difference between groups 3, 4 and 5. The ESS score showed worsening excessive daytime sleepiness from groups 1 to 5 with significant differences between groups 4 and 5. The burden of comorbidities in each group is presented in Figure 2.

### 3.2. Correlation Analyses

Both TST90% and TST80% correlated with age (ρ = 0.20 and ρ = 0.12), BMI (ρ = 0.51 and ρ = 0.44), SBP (ρ = 0.27 and ρ = 0.22), DBP (ρ = 0.19 and ρ = 0.13), HDL-C (ρ = −0.22 and ρ = −0.15), triglycerides (ρ = 0.23 and ρ = 0.18), CRP (ρ = 0.09 and ρ = 0.13) and ESS score (ρ = 0.32 and ρ = 0.29); however, only TST90% correlated with LDL-C (r = 0.10, all *p* < 0.05). There was no relationship between total cholesterol and TST90% or TST80% (*p* > 0.05).

### 3.3. Comparison of Diagnostic Performance of TST90% and TST80% to Detect Comorbidities

Both parameters were modestly predictive for OSA-related comorbidities. The performance of TST80% was not superior compared to TST90%. As a control, the performance of AHI was also plotted. The AUCs are summarised in Table 2.

According to the ROC analysis, AHI predicted increased 10-year cardiovascular risk the best, followed by TST90% and TST80% (Figure 3).

### 3.4. Adjusted Comparison of Groups 4 and 5

There was no difference between the two groups in the prevalence of comorbidities, cardiovascular risk or the SBP, DBP, CRP or lipid values, following adjustment for the relevant factors (all *p* > 0.05). The difference in ESS score between the two groups remained significant following adjustment (*p* = 0.02).

### 3.5. Comparison of Patients with Increased and Normal Cardiovascular Risk

Due to some missing data in 20 subjects, Framingham risk score was calculated in 777 volunteers. Comparing the two groups, the expected differences were significant for most parameters, except for DPB, CRP and TST80% (Table 3).

### 3.6. Sensitivity Analyses in Patients Who Had Polysomnography as a Diagnostic Test

Comparing clinical characteristics among the groups of patients in this subcohort, the results were similar to the whole analysis. Some post hoc intergroup differences became insignificant, but this could have been due to the lower number of subjects compared to the original dataset (Table 4).

Similarly, the diagnostic performance of TST80% was not superior to TST90% in detecting comorbidities and 10-year cardiovascular risk (Table 5).

## 4. Discussion

In this retrospective analysis of two cohort studies, we investigated the clinical utility of TST80% in patients with OSA. Although we found that patients with the most severe overnight hypoxaemia suffer from more severe daytime sleepiness, they did not have higher prevalence of cardiovascular and metabolic comorbidities or higher levels of blood pressure, CRP or more severe dyslipidaemia compared to patients with moderate overnight hypoxaemia. Consequently, the cardiovascular risk score was not higher in the most severe hypoxaemic group compared to the moderate group.

Nocturnal hypoxaemia is a hallmark of OSA and is a major factor linking OSA to its cardiovascular comorbidities. Whilst animal studies usually showed a dose–response relationship between the level of hypoxaemia and physiological variables, these studies also need to be interpreted with caution as animal response to very deep hypoxaemia may not necessarily be similar to humans [32]. In addition, hypoxaemia may also induce counterbalancing mechanisms. For instance, worsening hypoxaemia induced increasing glucose concentrations [22], but also unchanged insulin levels [22] and higher glucose transporter 1 expression [33], meaning that these do not necessarily lead to insulin resistance. Furthermore, whilst worsening hypoxaemia has led to increasing triglycerides, cholesterol levels were unchanged [20]. Finally, whilst oxygen tensions have an inverse relationship with catecholamine release [20], they also induce the expression of angiogenic factors such as the vascular endothelial growth factor, which has a vasodilator effect [33]. These mechanisms are further complicated in patients with OSA who show genetic, epigenetic and environmental (medications, smoking, diet, exercise) variations.

The relationship between the hypoxaemia indices and CRP was weak, and patients with more severe hypoxaemia did not demonstrate higher CRP levels. These results are in line with a previous study demonstrating that TST90%, but not TST80%, was related to inflammatory burden [11]. In addition, the Icelandic Sleep Apnea Cohort study did not find an independent association between time spent in hypoxaemia and CRP concentrations [34]. Although HIF-1α and inflammation are strongly interrelated at the molecular level [35], this cannot be directly translated in patients with OSA. For instance, sleep restriction may induce an anti-inflammatory response [36], and patients with very severe hypoxaemia might also have hypercapnia, which is known to decrease NF-κB activity [37]. Finally, patients with OSA may also have higher levels of anti-inflammatory proteins mitigating the inflammatory response [38].

The apnoea–hypopnea index, the marker used for the diagnosis and assessment of disease severity, has been frequently criticised, as it cannot fully capture the oxidative burden of OSA. Its value also depends on whether the former or the current AASM criteria were used for scoring, which makes comparisons between various studies difficult [39]. Similarly to AHI, TST90% and TST80% are readily available in sleep reports, but are less dependent on scoring. Their largest limitation is that disorders leading to nocturnal hypoxaemia other than OSA may influence their value. To avoid this bias, we excluded patients with these conditions in our study. As an alternative, it is possible to divide TST90% into acute desaturation-related and non-specific components using external software. Both were better related to cardiovascular mortality than the ODI; however, they did not outperform the overall TST90% [40]. TST90% was shown to better predict major cardiovascular events [12,13] and all-cause mortality compared to AHI [10,11,13] as well as cardiovascular mortality compared to ODI [40]. Although patients with more severe nocturnal hypoxaemia had an increased burden of comorbidities, TST90% or TST80% did not predict comorbidities or 10-year cardiovascular risk better than the AHI in our study. TST80% was associated with all-cause mortality, in contrast to TST90%, following adjustment on comorbidities, sleep parameters and inflammation [11]. Whilst our study had a case–control design, Smagula et al. managed to capture deaths in a cohort study. Of note, the excess in mortality was due to malignant rather than cardiovascular diseases [11]. Our results are in line with a recent study by Sutherland et al., who reported that TST90% was more strongly related to incidental cardiovascular disease than TST80% [27]. Our study extends the former one, as it also investigated individual components contributing to the cardiovascular risk.

The biggest limitation of our study is its cross-sectional nature; instead of capturing cardiovascular events, it only estimated the cardiovascular risk. Follow-up interventional (PAP or nocturnal oxygen) studies may better stratify patients based on TST80% vs. TST90%. Second, we did not have data on lifestyle factors, such as diet, alcohol consumption or exercise. Third, the diagnosis in some patients was based on polygraphy rather than polysomnography, which is an accepted diagnostic modality in uncomplicated cases [30]. For both TST80% and TST90%, the total sleep time was estimated based on changes in physiological variables and video analyses when polygraphy was attended. We believe that errors due to this estimation were minimal. However, as the scoring criteria for hypopnea is different for PSG than polygraphy, this could have led to errors when comparing and adjusting for AHI. To mitigate this, we performed sensitivity analyses in patients who had PSG as a diagnostic test. The results remained unchanged analysing only these patients, which validates our finding in the overall cohort. Fourth, we did not perform early morning blood gases. As discussed above, hypercapnia may counterbalance some consequences of hypoxaemia [37], and a higher proportion of patients with hypercapnia is plausible in more severe groups. Fifth, although comparisons were adjusted for significant confounders, the groups were not fully balanced for demographics. For instance, younger age in group 5 could have contributed to the lack of cardiovascular risk increase compared to group 4. Finally, the sample size in group 5 was low; therefore, post hoc comparisons with this group need to be interpreted with caution. The strengths of the study include the large sample size, detailed laboratory data and comprehensive characterisation of the patients.

In conclusion, the study suggests that TST80% has minimal clinical value compared to TST90% for estimating cardiovascular burden in patients with OSA. However, we cannot exclude that this parameter would not be valuable for association with malignant or mental health diseases, which were not evaluated. In addition, further studies are warranted to understand its value in all-cause mortality.

## Figures and Tables

**Figure 1 jcm-12-04205-f001:**
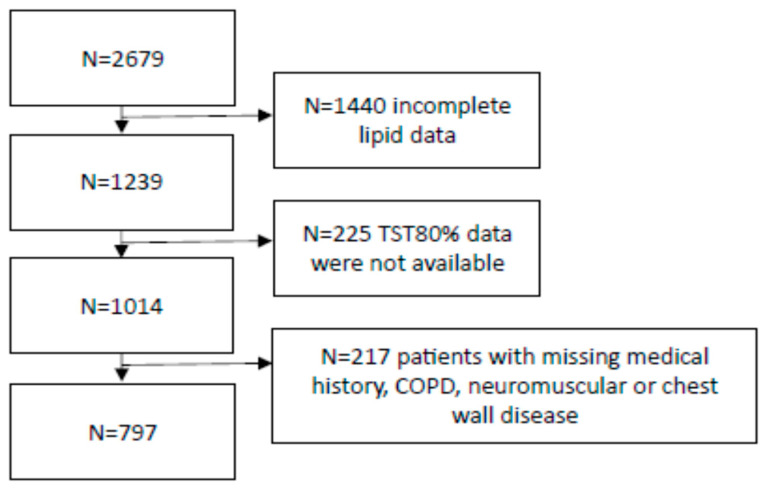
Algorithm for patient selection. TST80%—percentage of total sleep time spent with saturation below 80%; COPD—chronic obstructive pulmonary disease.

**Figure 2 jcm-12-04205-f002:**
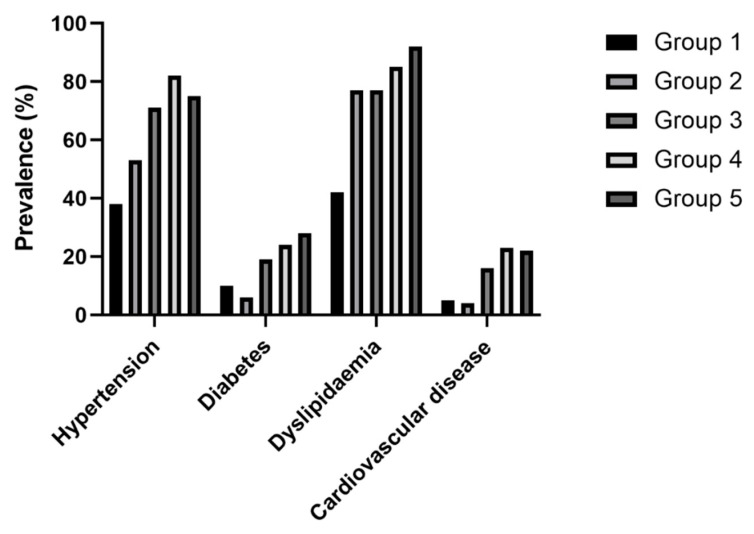
The burden of comorbidities in each group. The prevalence of hypertension, diabetes, dyslipidaemia and cardiovascular disease is plotted in each group. Group 1—no OSA; Group 2—non-hypoxaemic OSA (AHI ≥ 5/h, but TST90% = 0%); Group 3—minimally hypoxaemic OSA (AHI ≥ 5/h, TST90% 0–10%); Group 4—moderately hypoxaemic OSA (AHI ≥ 5/h, TST90% ≥ 10%, but TST80% < 10%) and Group 5—severely hypoxaemic OSA (AHI ≥ 5/h, and TST80% ≥ 10%). OSA—obstructive sleep apnoea; AHI—apnoea–hypopnea index; TST90%—percentage of total sleep time spent with saturation below 90%; TST80%—percentage of total sleep time spent with saturation below 80%.

**Figure 3 jcm-12-04205-f003:**
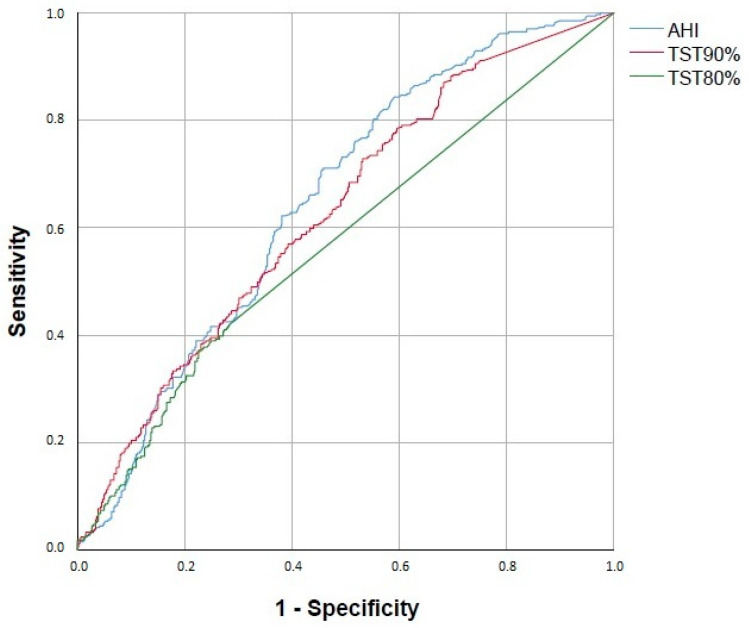
Receiver operating characteristics curve analysis to predict 10-year cardiovascular risk. AHI—apnoea–hypopnea index; TST90%—percentage of total sleep time spent with saturation below 90%; TST80%—percentage of total sleep time spent with saturation below 80%.

**Table 1 jcm-12-04205-t001:** Comparison of the groups.

	Group 1 (*n* = 92)	Group 2 (*n* = 79)	Group 3 (*n* = 396)	Group 4 (*n* = 194)	Group 5 (*n* = 36)	*p* Value
Age (years)	46 ± 15	49 ± 12	55 ± 12 ^#¶^	56 ± 11 ^#¶^	49 ± 10 ^µ^	<0.001
Gender (males%)	27	62	67	74	83	<0.001
BMI (kg/m^2^)	25 ± 5	29 ± 5 ^#^	32 ± 6 ^#¶^	36 ± 7 ^#¶@^	39 ± 7 ^#¶@^	<0.001
Current smokers (%)	5	23	19	15	33	0.001
Hypertension (%)	38	53	71	82	75	<0.001
Diabetes (%)	10	6	19	24	28	<0.001
Dyslipidaemia (%)	42	77	77	85	92	<0.001
Cardiovascular disease (%)	5	4	16	23	22	<0.001
Lipid-lowering medication (%)	5	25	37	43	25	<0.001
SBP (mmHg)	124 ± 17	130 ± 13	135 ± 14 ^#^	138 ± 14 ^#¶@^	137 ± 15 ^#^	<0.001
DBP (mmHg)	77 ± 10	80 ± 9	82 ± 10 ^#^	85 ± 11 ^#¶@^	83 ± 10 ^#^	<0.001
Total cholesterol (mmol/L)	5.4 ± 1.0	5.3 ± 1.1	5.3 ± 1.2	5.3 ± 1.1	5.7 ± 1.2	0.27
LDL-C (mmol/L)	3.1 ± 1.0	3.3 ± 1	3.3 ± 1.0	3.4 ± 1.0	3.7 ± 1.0 ^#^	0.04
HDL-C (mmol/L)	1.7 ± 0.6	1.2 ± 0.3 ^#^	1.2 ± 0.4 ^#^	1.2 ± 0.3 ^#^	1.1 ± 0.2 ^#^	<0.001
Triglycerides (mmol/L)	1.3 ± 0.7	1.7 ± 0.7	1.8 ± 0.9 ^#^	2.1 ± 1.2 ^#¶@^	2.0 ± 0.6 ^#^	<0.001
CRP (mg/L)	4.5 ± 12.6	2.2 ± 7.1	2.1 ± 3.8 ^#^	2.9 ± 5.7	2.0 ± 2.6	0.03
Framingham risk score	3.3 ± 5.4	7.0 ± 7.6 ^#^	9.7 ± 7.9 ^#¶^	11.4 ± 7.3 ^#¶^	11.3 ± 8.6 ^#¶^	<0.001
Patients at >10% cardiovascular risk (%)	12	31	45	56	52	<0.001
AHI (1/h)	2.4 ± 1.4	18.7 ± 9.8 ^#^	28.9 ± 17.9 ^#¶^	48.7 ± 23.0 ^#¶@^	76.7 ± 16.4 ^#¶@µ^	<0.001
ODI (1/h)	1.2 ± 1.1	14.1 ± 9.1 ^#^	24.1 ± 17.4 ^#¶^	45.0 ± 24.4 ^#¶@^	78.5 ± 16.9 ^#¶@µ^	<0.001
TST90%	0 ± 0	0 ± 0	2.8 ± 2.8	28.5 ± 17.3 ^#¶@^	63.6 ± 16.6 ^#¶@µ^	<0.001
TST80%	0 ± 0	0 ± 0	0.1 ± 0.3	1.6 ± 2.1 ^#¶@^	27.3 ± 17.5 ^#¶@µ^	<0.001
ESS score	6.1 ± 3.4	6.7 ± 4.1	7.9 ± 4.0 ^#^	9.8 ± 4.1 ^#¶@^	12.3 ± 3.4 ^#¶@µ^	<0.001

AHI—apnoea–hypopnoea index; BMI—body mass index; CRP—C-reactive protein; DBP—diastolic blood pressure; ESS—Epworth Sleepiness Scale; HDL-C—high-density lipoprotein cholesterol; LDL-C—low-density lipoprotein cholesterol; ODI—oxygen desaturation index; SBP—systolic blood pressure; TST80%—the percentage of total sleep time spent with saturation below 80%; TST90%—the percentage of total sleep time spent with saturation below 90%.^#^
*p* < 0.05 vs. group 1; ^¶^ *p* < 0.05 vs. group 2; ^@^ *p* < 0.05 vs. group 3; ^µ^ *p* < 0.05 vs. group 4.

**Table 2 jcm-12-04205-t002:** Comparison of areas under the receiver operating characteristics curves.

	Hypertension	Diabetes	Dyslipidaemia	Cardiovascular Disease
TST90%	0.67/0.63–0.71/	0.62/0.58–0.67/	0.63/0.58–0.67/	0.65/0.61–0.70/
TST80%	0.60/0.56–0.64/	0.58/0.53–0.63/	0.59/0.55–0.64/	0.61/0.55–0.66/
AHI	0.67/0.63–0.71/	0.60/0.55–0.65/	0.67/0.63–0.72/	0.61/0.56–0.66/

AHI—apnoea–hypopnoea index; TST80%—the percentage of total sleep time spent with saturation below 80%; TST90%—the percentage of total sleep time spent with saturation below 90%.

**Table 3 jcm-12-04205-t003:** Comparison of patients with increased and normal cardiovascular risk.

	Normal Cardiovascular Risk (*n* = 440)	Increased Cardiovascular Risk (*n* = 337)	*p* Value
Age (years)	49 ± 12	60 ± 10	<0.001
Gender (males%)	48	88	<0.001
BMI (kg/m^2^)	32 ± 7	33 ± 6	<0.001
Current Smoker (%)	10	28	<0.001
Hypertension (%)	56	87	<0.001
Diabetes (%)	13	26	<0.001
Dyslipidaemia (%)	70	83	<0.001
Cardiovascular disease (%)	10	24	<0.001
SBP (mmHg)	131 ± 14	139 ± 14	<0.001
DBP (mmHg)	81 ± 11	82 ± 9	0.131
Total cholesterol (mmol/L)	5.2 ± 1.1	5.5 ± 1.2	<0.001
LDL-C (mmol/L)	3.2 ± 1.0	3.6 ± 1.1	<0.001
HDL-C (mmol/L)	1.3 ± 0.5	1.1 ± 0.3	<0.001
Triglycerides (mmol/L)	1.7 ± 1.0	2.0 ± 1.0	<0.001
CRP (mg/L)	2.4 ± 5.8	2.7 ± 6.4	0.463
AHI (1/h)	27.8 ± 23.9	38.4 ± 23.5	<0.001
ODI (1/h)	24.1 ± 23.9	34.4 ± 24.7	<0.001
TST90%	8.9 ± 16.6	14.8 ± 20.9	<0.001
TST80%	1.3 ± 5.2	2.2 ± 8.6	0.061
ESS score	7.9 ± 4.2	8.9 ± 4.2	0.001

AHI—apnoea–hypopnoea index; BMI—body mass index; CRP—C-reactive protein; DBP—diastolic blood pressure; ESS—Epworth Sleepiness Scale; HDL-C—high-density lipoprotein cholesterol; LDL-C—low-density lipoprotein cholesterol; ODI—oxygen desaturation index; SBP—systolic blood pressure; TST80%—the percentage of total sleep time spent with saturation below 80%; TST90%—the percentage of total sleep time spent with saturation below 90%.

**Table 4 jcm-12-04205-t004:** Comparison of clinical characteristics in patients who had polysomnography as a diagnostic test.

	Group 1 (*n* = 67)	Group 2 (*n* = 51)	Group 3 (*n* = 295)	Group 4 (*n* = 155)	Group 5 (*n* = 30)	*p* Value
Age (years)	45 ± 16	48 ± 12	54 ± 12 ^#¶^	56 ± 11 ^#¶^	49 ± 9 ^µ^	<0.001
Gender (males%)	24	65	68	76	83	<0.001
BMI (kg/m^2^)	25 ± 4	29 ± 6 ^#^	32 ± 6 ^#¶^	36 ± 6 ^#¶@^	40 ± 7 ^#¶@µ^	<0.001
Current smokers (%)	6	22	18	15	37	0.001
Hypertension (%)	34	47	68	83	73	<0.001
Diabetes (%)	7	6	17	26	27	<0.001
Dyslipidaemia (%)	46	73	76	85	93	<0.001
Cardiovascular disease (%)	4	2	15	22	23	<0.001
Lipid-lowering medication (%)	7	22	38	47	23	<0.001
SBP (mmHg)	124 ± 14	129 ± 13	135 ± 14 ^#^	138 ± 14 ^#¶^	137 ± 15 ^#^	<0.001
DBP (mmHg)	77 ± 10	80 ± 9	81 ± 10 ^#^	85 ± 11 ^#¶@^	82 ± 10	<0.001
Total cholesterol (mmol/L)	5.3 ± 1.0	5.3 ± 1.2	5.2 ± 1.1	5.3 ± 1.1	5.8 ± 1.2	0.125
LDL-C (mmol/L)	3.0 ± 1.0	3.2 ± 1.1	3.3 ± 1.0	3.5 ± 1.0 ^#^	3.8 ± 0.9 ^#@^	<0.001
HDL-C (mmol/L)	1.7 ± 0.6	1.3 ± 0.4 ^#^	1.2 ± 0.5 ^#^	1.2 ± 0.2 ^#^	1.1 ± 0.2 ^#^	<0.001
Triglycerides (mmol/L)	1.3 ± 0.8	1.6 ± 0.5	1.8 ± 0.9 ^#^	2.1 ± 1.2 ^#¶^	2.0 ± 0.6 ^#^	<0.001
CRP (mg/L)	4.9 ± 14.0	2.4 ± 8.5	2.0 ± 4.1 ^#^	2.2 ± 3.7	1.7 ± 2.7	0.044
Framingham risk score	2.3 ± 3.9	6.5 ± 7.7 ^#^	9.3 ± 7.8 ^#^	11.4 ± 7.1 ^#¶@^	10.6 ± 8.5 ^#^	<0.001
Patients at >10% cardiovascular risk (%)	5	28	43	56	47	<0.001
AHI (1/h)	2.5 ± 1.4	19.1 ± 10.6 ^#^	28.0 ± 17.5 ^#¶^	48.2 ± 22.6 ^#¶@^	77.0 ± 15.4 ^#¶@µ^	<0.001
ODI (1/h)	1.1 ± 1.0	13.4 ± 10.0 ^#^	22.3 ± 15.8 ^#¶^	44.7 ± 24.7 ^#¶@^	79.2 ± 15.6 ^#¶@µ^	<0.001
TST90%	0 ± 0	0 ± 0	3.0 ± 2.8	30.6 ± 18.1 ^#¶@^	65.8 ± 16.8 ^#¶@µ^	<0.001
TST80%	0 ± 0	0 ± 0	0.1 ± 0.3	1.6 ± 2.1 ^#¶@^	28.7 ± 18.7 ^#¶@µ^	<0.001
ESS score	6.1 ± 3.4	6.7 ± 4.1	7.9 ± 4.0 ^#^	9.8 ± 4.1 ^#¶@^	12.3 ± 3.4 ^#¶@µ^	<0.001

AHI—apnoea–hypopnoea index; BMI—body mass index; CRP—C-reactive protein; DBP—diastolic blood pressure; ESS—Epworth Sleepiness Scale; HDL-C—high-density lipoprotein cholesterol; LDL-C—low-density lipoprotein cholesterol; ODI—oxygen desaturation index; SBP—systolic blood pressure; TST80%—the percentage of total sleep time spent with saturation below 80%; TST90%—the percentage of total sleep time spent with saturation below 90%. ^#^
*p* < 0.05 vs. group 1; ^¶^ *p* < 0.05 vs. group 2; ^@^ *p* < 0.05 vs. group 3; ^µ^ *p* < 0.05 vs. group 4.

**Table 5 jcm-12-04205-t005:** Comparison of area under the receiver operating characteristics curves in patients who had polysomnography as a diagnostic test.

	Hypertension	Diabetes	Dyslipidaemia	Cardiovascular Disease	10 Year Cardiovascular Risk
TST90%	0.68/0.63–0.73/	0.64/0.59–0.70/	0.67/0.62–0.72/	0.67/0.61–0.72/	0.64/0.60–0.69/
TST80%	0.61/0.57–0.66/	0.61/0.53–0.67/	0.60/0.55–0.66/	0.63/0.57–0.70/	0.59/0.54–0.64/
AHI	0.67/0.62–0.72/	0.62/0.55–0.65/	0.67/0.62–0.72/	0.62/0.56–0.68/	0.66/0.62–0.71/

AHI—apnoea–hypopnoea index; TST80%—the percentage of total sleep time spent with saturation below 80%; TST90%—the percentage of total sleep time spent with saturation below 90%.

## Data Availability

The datasets analysed during the current study are not publicly available due general data protection policy, but are available from the corresponding author on reasonable request.

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
