# Peer review of "Time Spent with Saturation below 80% versus 90% in Patients with Obstructive Sleep Apnoea"

_jcm, 2023, doi:10.3390/jcm12134205_

Round 1

Reviewer 1 Report

Were there any patients with OHS (obesity hypoventilation syndrome) at the study populatıon?

 When you were grouping OSA patients, why did you use 90%TST < 10%?

for compairing the importance of cardiovascular risk factors, I wish authors did not use that some parameteres that were obtained the different diagnostic tools such as PSG, POLYGRAPH at the same study.

Author Response

Comment: Were there any patients with OHS (obesity hypoventilation syndrome) at the study population?

Response: Thank you for raising this valid point. Unfortunately, blood gases were not performed. It is likely that more patients in more severe groups could have OHS as a comorbidity. Hypercapnia could counterbalance some mechanisms of hypoxaemia therefore the lack of difference in the primary outcomes could have been due to this error. We expanded the limitation with this information.

Comment: When you were grouping OSA patients, why did you use 90%TST < 10%?

Response: We used this cut off from a recent study on a large group of patients with OSA. We have expanded the Introduction. Please, see 2nd paragraph of Introduction and Methods.

Comment: For comparing the importance of cardiovascular risk factors, I wish authors did not use that some parameteres that were obtained the different diagnostic tools such as PSG, POLYGRAPH at the same study.

Response: Thank you for your comment. We performed sensitivity analyses in patients who had polysomnography as a diagnostic test. The results remained to be similar to those in the original.

Reviewer 2 Report

This is an interesting study addressing whether or not severe hypoxemia in terms of time spent below 80% oxygen saturation (TST80%) is superior to TST90% with regard to cardiovascular risk in patients with OSA, and the answer is no.

Though the research question is simple, the paper gives insights for a practical approach for determining cardiovascular risk based on the comparison of AHI with available oxygenation indices provided from the sleep recordings.

I have some concerns regarding the data presentation and conclusions drawn from the results:

1) The lack of additional value of TST80% with regard to cardiovascular risk may be related with the small sample size (n=36) of this subgroup compared with the larger sample size of TST90% (n=194). This should be acknowledged as one of the study limitations.

2) Not all patients underwent PSG so time spent below 80% vs 90% might be misleading due to lack of information about the total sleep time for the polygraphic measurements.

3) A similar limitation is also valid for AHI provided from PSGs vs polygraphic measurements since the first one includes >=3% desaturations and arousals whereas the latter is based on desaturations only. This should also be added as a limitation.

4) A separate sensitivity analysis should be done based on PSG results only in order to see if the results are consistent.

5) Patients in the TST80% subgroup were significantly younger than the TST90% group, which may also have contributed to the lack of dose-response relationship between the severity of desaturations and the cardiovascular risk.

6) Clarify if smoking refers to current smoking at the time of the sleep studies or if it also includes former smokers.

7) Clarify the Framingham risk score in the Methods section.

8) In this cohort, AHI gives the best risk score compared to the desaturation indices, which might be due to confounding effect of polygraphic investigations that are based on desaturations but not arousals. Comparisons should be done only within the PSG group.

Can be improved.

Author Response

Comment: 1) The lack of additional value of TST80% with regard to cardiovascular risk may be related with the small sample size (n=36) of this subgroup compared with the larger sample size of TST90% (n=194). This should be acknowledged as one of the study limitations.

Response: Thank you. We have expanded the limitation section with this comment.

Comment: 2) Not all patients underwent PSG so time spent below 80% vs 90% might be misleading due to lack of information about the total sleep time for the polygraphic measurements.

Response: We agree with the comment. In the revised manuscript, we performed sensitivity analyses in patients who had PSG as a diagnostic test. The results in these patients were similar to those in the whole group.

Comment: 3) A similar limitation is also valid for AHI provided from PSGs vs polygraphic measurements since the first one includes >=3% desaturations and arousals whereas the latter is based on desaturations only. This should also be added as a limitation.

Response: We agree with the comment. The limitation section was expanded, and a sensitivity analysis was performed in patients who had PSG as a diagnostic test.

Comment: 4) A separate sensitivity analysis should be done based on PSG results only in order to see if the results are consistent.

Response: Based on your comment, a sensitivity analysis was performed only in patients who had PSG as a diagnostic test. The results were similar in these patients to the whole cohort.

Comment: 5) Patients in the TST80% subgroup were significantly younger than the TST90% group, which may also have contributed to the lack of dose-response relationship between the severity of desaturations and the cardiovascular risk.

Response: Thank you. Statistical analyses were adjusted for age, however, we acknowledged your concern as a potential limitation in the revised manuscript.

Comment: 6) Clarify if smoking refers to current smoking at the time of the sleep studies or if it also includes former smokers.

Response: Thank you for highlighting this issue which identified an error regarding the definition of ex-smoker between the sites. According to the Framingham study “Persons who smoked regularly during the previous 12 months were classified as smokers”. We have updated the database to reflect on this definition and recalculated the results (Please, see amendments in the results in track changes). We have also clarified the definition in the Methods.

Comment: 7) Clarify the Framingham risk score in the Methods section.

Response: Thank you. We have now better described the Framingham risk score in the Methods.

Comment: 8) In this cohort, AHI gives the best risk score compared to the desaturation indices, which might be due to confounding effect of polygraphic investigations that are based on desaturations but not arousals. Comparisons should be done only within the PSG group.

Response: We performed sensitivity analysis only in the PSG group. AHI remained the best predictor for the 10-year cardiovascular risk.

Round 2

Reviewer 2 Report

Thank for the revision. The paper seems to be ımproved.

Acceptable